# Improving the Performance of Polydimethylsiloxane-Based Triboelectric Nanogenerators by Introducing CdS Particles into the Polydimethylsiloxane Layer

**DOI:** 10.3390/nano13222943

**Published:** 2023-11-14

**Authors:** Jianbin Mao, Soonmin Seo

**Affiliations:** Department of Bionano Technology, Gachon University, Seongnam 13120, Republic of Korea; maojianbin@gachon.ac.kr

**Keywords:** metal chalcogenides, CdS, TENG, output performance, nanomaterials

## Abstract

Energy harvesting and power generation technologies hold significant potential for meeting future energy demands and improving environmental sustainability. A triboelectric nanogenerator (TENG), which harnesses energy from the surrounding environment, has garnered significant attention as a promising and sustainable power source applicable in various fields. In this study, we present a technique to improve the triboelectric performance of a PDMS-based TENG by incorporating nanostructured cadmium sulfide (N-CdS). This study investigates the utilization of CdS nanomaterials in TENG production, where mechanical energy is converted into electrical energy. We conducted a comparative analysis of TENGs utilizing N-CdS/PDMS, commercial CdS/PDMS (C-CdS/PDMS), and pure PDMS substrates. The N-CdS/PDMS substrates demonstrated superior triboelectric performance compared to TENG devices based on pure PDMS and C-CdS/PDMS. The triboelectric open-circuit voltage (V_oc_) and short-circuit current (I_sc_) of the N-CdS/PDMS-based TENG device were approximately 236 V and 17.4 µA, respectively, when operated at a 2 Hz frequency. These values were approximately 3 times and 2.5 times higher, respectively, compared to the pure PDMS-based TENGs. They were further studied in detail to understand the effect of different parameters such as contact–separation frequency and contact force on the TENGs’ operation. The stability of the TENG devices was studied, and their potential to be integrated into self-powered smart textiles as power sources was demonstrated.

## 1. Introduction

The topic of energy sources has long been a subject of significant focus particularly in relation to the utilization and scarcity of fossil fuels [1]. Therefore, energy harvesting is crucial for our daily lives and is receiving significant attention. A TENG is a well-designed method capable of converting mechanical energy from the environment into electrical energy through the coupling of triboelectrification and electrostatic induction [2]. According to the relevant literature, a TENG has the capability to convert mechanical energy from various external forces, such as wind, water, and even body motion, into electrical energy [3,4,5]. Since their introduction and invention by Wang’s group in 2012, TENGs have been recognized as a promising method for energy harvesting and a sustainable renewable resource [6]. TENGs offer numerous advantages when compared to other methods of energy acquisition. These benefits include high-efficiency power conversion, a simple structure, ease of fabrication, low cost, durability, scalability, and the ability to harvest energy from low-frequency, irregular input sources [7,8]. Overall, due to these advantages, TENGs are becoming increasingly integrated into people’s daily lives.

However, increasing the power output of TENGs is essential to meet their various practical applications [9], as the current output power of TENGs remains significantly below the ideal value. Moreover, there are various factors that can compromise the efficiency of TENGs [10,11]. Hence, conducting comprehensive research to enhance performance and address various challenges faced by TENGs is crucial to advance their practical applications. Selecting and modifying materials, as well as employing surface treatment techniques, are crucial approaches for effectively enhancing a TENG’s power output. For example, our group deposited Galinstan onto a polydimethylsiloxane (PDMS) substrate and designed a Galinstan-PDMS-based TENG, resulting in a significant increase in output power [12,13,14]. Kavarthapu et al. discovered that the output performance of TENG could be enhanced through the modulation of the loading concentration of ZnSnO_3_ nanoparticles (NPs) in polyvinylidene fluoride hexafluoropropylene (PVDF-HFP) fibrous films [15]. Li et al. suggest the utilization of ordered mesoporous SiO_2_ (OMS) nanoparticles, characterized by a substantial specific surface area, as efficient sites for body charge storage within OMS-PDMS to augment the output performance of TENGs [16]. The above studies highlight the significance of material modification and selection.

Metal chalcogenides are intriguing materials that have attracted significant attention [17]. In recent years, metal chalcogenides have found extensive applications, including batteries, photovoltaics, nanoelectronics, nanotribology, catalysis, and more [18,19,20,21,22]. Furthermore, metal chalcogenides have also entered the field of energy harvesting. For example, because ZnS possesses piezoelectric and triboelectric properties, Mishra and colleagues introduced ZnS nanosheets onto a surface-modified aluminum substrate [23]. Their report demonstrated that ZnS can significantly enhance the charge transfer rate between two tribolayers. CdS, as a representative member of metal chalcogenides and one of the most widely used II-VI semiconductors, has been extensively studied owing to its captivating physical and chemical properties [24]. CdS exists in two crystalline forms: hexagonal (wurtzite) and cubic (zinc blende), with the ability to produce CdS films in both phases [25]. CdS is categorized as an n-type semiconductor with a direct band gap of 2.42 eV in its cubic structure and 2.57 eV in its hexagonal form [26]. CdS offers numerous advantages, including a high absorption coefficient (>10^4^ cm^−1^), excellent photoelectric properties, strong reflectance in the infrared range, high transmittance in the visible spectrum, low work function, superior electron affinity, high carrier concentration (10^16^–10^18^ cm^−3^), high carrier mobility (0.1–10 cm^2^ V^−1^ s^−1^), high electron mobility (300 cm^2^ V^−1^ s^−1^), low resistivity, and exceptional thermochemical and electrochemical stability [27,28,29,30]. CdS is widely used in photosensors, photovoltaic devices, and solar cells [31]. Additionally, owing to the semiconductor and piezoelectric characteristics of CdS, it is employed in energy conversion and harvesting. Furthermore, there have been reports of piezoelectric nanogenerators (PENGs) incorporating CdS [32]. However, the use of CdS as a material for TENGs remains relatively underreported.

The polymer film PDMS is commonly used as a negative triboelectric layer in the fabrication of TENGs. In this study, we introduced a novel approach by utilizing the triboelectrification of a dielectric composite film composed of CdS and PDMS, resulting in the creation of an easily manufacturable and cost-effective TENG for the first time. The CdS/PDMS film was combined with a copper film as the triboelectric contact material, and the resulting CdS/PDMS-based TENG operated in a contact–separation mode. We systematically investigated the impact of introducing CdS on the output performance of the TENG by measuring the voltage and current when CdS was incorporated into the PDMS film. Initially, through a comparison with pure PDMS, we observed a significant improvement in the output performance of the TENG following the introduction of CdS. This enhancement primarily stemmed from the augmented contact area, accelerated charge transfer, and improved conductivity facilitated by the introduction of CdS. Our preparation involved the transformation of commercially available micron-level CdS into nanostructured CdS. The test results revealed that the performance of the nanostructured N-CdS was further amplified upon integration with PDMS, demonstrating a maximum output voltage of 236 V and a current of 17.4 μA. Furthermore, simulation under real circuit conditions, utilizing an external load resistor, indicated an output energy density of 4 μW/m^2^ based on N-CdS/TENG. Finally, our TENG successfully illuminated 17 LED lights concurrently, underscoring its potential for practical applications.

## 2. Materials and Methods

### 2.1. Preparation of Nanostructured CdS

CdS nanomaterials were synthesized using commercial CdS (99.999% purity, Alfa Aesar, Tewksbury, MA, USA) through an ultrasonication method. Specifically, 4 g of commercial CdS was dispersed in deionized water. Subsequently, the solution was ultrasonicated for 16 h (ultrasound equipment: SONICS-VCX500, SONICS, Newtown, CT, USA). Additionally, the set power is 200 W. After ultrasonication, the CdS solution was allowed to settle for 2 h. The upper layer was then centrifuged at 3000 rpm for 40 min (centrifuge: UNIVERSAL 320R, Hettich, Beverly, MA, USA). Subsequently, the CdS nanomaterials were collected and dried in an oven.

### 2.2. Fabrication of TENGs

The CdS-based TENG fabrication process is illustrated in Figure 1. First, the PDMS (Sylgard 184, Dow Corning, Midland, MI, USA) was prepared by mixing base and curing agents at the weight percentage of 10/1 (*w*/*w*), and then CdS (commercial CdS, and nanostructured CdS) was dispersed into the PDMS elastomer at the concentration of 0.6 wt%. After mixing thoroughly, the mixture was placed in a container and subjected to a vacuum for 0.5 to 1 h. This step aimed to remove any dissolved air from the mixture. The solution was then used to fabricate a 200 μm thick layer on a polyethylene terephthalate (PET) substrate via a bar coating using a 4-side applicator. The PET tape was washed several times with water and ethanol. A 200 μm wire-wound rod was used to deposit the CdS/PDMS-based thin films. After depositing the CdS/PDMS on the PET, the composite thin membrane was dried in an oven at constant temperature of 60 °C for approximately 2 h. After the curing step, the cured composite thin film with CdS was sliced to 2.5 × 2.5 cm^2^. The TENG device was fabricated using a simple two-electrode system. For the bottom electrode, copper foil was utilized as the electrode with the thin composite film (CdS/PDMS based PET tape) directly attached to it. The upper electrode, made of copper, functioned as both the charge-collecting electrode (CCE) and the frictional electrode (FE). Electrical contact was established through Cu wire connecting the bottom and upper electrodes. Subsequently, the TENG was constructed.

### 2.3. Characterization

Field emission scanning electron microscope (FESEM, JSM-7500F, JEOL, Tokyo, Japan) analysis was conducted. UV-visible absorption spectra were obtained using a Shimadzu UV-2550 spectrometer. FT-IR spectra were obtained using a Thermo Fisher Scientific (iS50) FT-IR spectrometer.

### 2.4. Measurement of TENGs

To investigate the enhanced triboelectricity generation due to the nanostructured CdS particle layer, we measured the voltage and current of the device. The open circuit voltage was measured using an oscilloscope (MDO34, Tektronix, Beaverton, OR, USA), while an electrometer (Keithley-6514, Keithley, Cleveland, OH, USA) quantified the short-circuit current. A linear motor served as a controlled vibration source during electrical performance tests, with the motor integrated with a force sensor to measure applied force on the TENG.

## 3. Results and Discussion

To investigate the surface morphology and size of CdS nanomaterials prepared using ultrasonication, we conducted a study using FE-SEM and compared them to C-CdS (referred to as original CdS), as depicted in Figure 2. Figure 2a,b display the morphology and size of the C-CdS. From Figure 2a, it is evident that C-CdS predominantly exhibits a blocky morphology reminiscent of variously shaped and sized gravel. We illustrate the size distribution of C-CdS through size measurements. In Figure 2b, the size distribution of C-CdS ranges from 1 to 10 μm. Based on our measurements, C-CdS with a size between 2 and 5 μm constitutes the majority, indicating that the C-CdS is at the micron level in size. Figure 2c–f depict the morphology and size distribution of N-CdS obtained through ultrasonic treatment of the C-CdS. A comparison of Figure 2c with Figure 2a reveals a substantial reduction in the size of CdS. This result suggests that with a specific ultrasonic duration, the CdS size can be diminished, which is consistent with prior research findings [33]. Further size analysis indicates that cadmium sulfide nanomaterials primarily range from a few hundred nanometers to 2.5 μm with their maximum size considerably smaller than the original CdS. Furthermore, the result confirms that the ultrasonic method can produce sub-micrometer-sized CdS. While the size distribution diagram reveals some CdS exceeding 1 μm, this could be attributed to uneven ultrasonication or timing issues. We anticipate that extending the ultrasonication duration will yield smaller CdS. In Figure 2e,f, we discern a clearer view. Here, C-CdS treated by ultrasonication predominantly exhibits particle agglomerates of varying sizes primarily within the range of several hundred nanometers. The variations in size are likely due to differences in the original CdS, and the particle agglomerates morphology may be linked to the layered structure of CdS and the bulk morphology of C-CdS, as the ultrasonic process may resemble a stripping process. Furthermore, however, in comparison to the C-CdS, N-CdS exhibits a tendency to stack, primarily attributed to the size reduction, which promotes material agglomeration. In summary, the findings above demonstrate the successful transformation of micron-level CdS into nanoscale CdS through a straightforward ultrasonication process.

Next, we examined the optical properties of CdS nanomaterials. Figure 3a exhibits the UV-visible spectrum of CdS nanomaterials and commercial CdS sample obtained through the recording of absorption spectra using DI water as a reference. Consistent with prior findings, a distinct peak at 290 nm was observed, confirming its association with CdS nanomaterials [34]. This result further attests to the successful synthesis of nanostructured N-CdS via a straightforward ultrasonication method and highlights the strong electron confinement effect in N-CdS. Moreover, we employed FT-IR analysis to investigate CdS, PDMS, and CdS/PDMS. Figure 3b does not display the anticipated peak corresponding to CdS. Upon comparing the primary absorption peaks of PDMS before and after the introduction of CdS, no noticeable disparity was observed between the two. In the infrared spectrum (Figure 3b), the absorption peak near 2964 cm^−1^ is attributed to the C-H stretching vibration of CH_3_, and the peak near 1259 cm^−1^ arises from the symmetrical bending of CH_3_ in Si-CH_3_. The appearance of peaks near 1067 cm^−1^ and 1010 cm^−1^ can be linked to the formation of Si-O-Si bonds, while the 787 cm^−1^ absorption peak emerges due to the swinging motion of CH_3_ in Si-CH_3_. These results indicate that the introduction of CdS did not significantly alter the primary chemical bonds of PDMS, which was possibly due to the minimal quantity of CdS introduced.

In this work, CdS was utilized as a co-triboelectric layer material in the TENG. Figure 4 illustrates a schematic diagram depicting the operating mechanism of the CdS-PDMS-based TENG. When no external force is applied, the TENG is in its initial state (Figure 4-i), where the top and bottom sides are separated. When a force is applied to the TENG (Figure 4-ii), the top and bottom sides come into contact, inducing negative and positive electrostatic charges on the surfaces the upper electrode and the lower electrode, respectively. The separation of the Cu layer and the CdS/PDMS composites results in the generation of negative and positive electrostatic charges on the surfaces of both the upper electrode layer and the lower electrode layer, respectively, through electrostatic induction. Consequently, current flows from the top to the bottom side of the TENG (Figure 4-iii). Once the electrostatic charges reach equilibrium, no current flows in the external circuit (Figure 4-iv). Moreover, when a force is applied to the TENG, reverse current flows from the bottom to the top side (Figure 4-v). Thus, current continuously flows between the bottom and top sides of the TENG as a force is continuously applied and removed.

A TENG with an area of 2.5 cm × 2.5 cm was used to measure triboelectric energy. The spacing between the friction layers was 45 mm, and it operated at a frequency of 2 Hz with an applied force of 5 N, as shown in Figure 5a. Initially, we altered the arrangement of the interconnection between the friction layers to assess any impact on the output voltage and current. The results are depicted in Figure 5b,c. It was observed that changing the interconnection direction solely affected the polarity of the voltage and current while leaving their magnitudes unaffected. This finding underscores the independence of the output voltage and current from the interconnection direction, which aligns with earlier research findings [35]. The performance of the TENG primarily depends on the spacing between the friction layers, the dielectric constant, the design structure, the friction frequency, and other relevant factors. Different friction layers, including PDMS, C-CdS/PDMS, and N-CdS/PDMS, were used to investigate and compare their output properties within the TENG system. As shown in Figure 5d, the voltage output when using PDMS as the friction layer measured 80 V, while C-CdS/PDMS and N-CdS/PDMS produced significantly higher output voltages of 204 V and 236 V, respectively, corresponding to approximately 2.6 times and 3 times that of pure PDMS. Likewise, Figure 5e illustrates that the current output with PDMS as the friction layer was 7.2 µA, while using C-CdS/PDMS and N-CdS/PDMS as the friction layer resulted in output currents of 16 µA and 17.2 µA, respectively, representing 2.2 times and 2.4 times the original PDMS output. Evidently, incorporating CdS into the PDMS as the friction layer markedly amplifies both the output voltage and current of the TENG. These results substantiate the significant performance enhancement achieved by the introduction of CdS into the PDMS-based TENG system. The superior output performance of the CdS/PDMS-based TENG can be attributed to the following reasons: (i) the introduction of CdS significantly enhances the charge transfer rate and conductivity; and (ii) CdS provides a rough surface, offering a larger contact area compared to pure PDMS. CdS is a piezoelectric material, as highlighted in the Introduction. Combining the piezoelectric and triboelectric effects can significantly enhance the performance of TENG [36]. When a mechanical force is applied, the triboelectric material PDMS, containing piezoelectric properties of CdS, undergoes deformation, leading to an immediate manifestation of piezoelectricity. The supplementary surface charges resulting from the piezoelectric effect of CdS reinforce the triboelectric charge during simultaneous contact electrification, ultimately boosting the device’s electrical outputs. For TENGs utilizing N-CdS/PDMS fabricated from nanostructured CdS as the friction layer, the observed output performance surpasses that of C-CdS/PDMS derived from micron-sized CdS. This enhancement can be primarily attributed to the smaller size of the nanoscale N-CdS, as evident from the SEM image in Figure 2. Additionally, the evaluation of their transferred charge densities reveals that N-CdS/PDMS exhibits the highest transferred charge density, measuring 55 µC/m2, as shown in Figure 5f. This outcome is a result of the accelerated charge transfer by the nanostructure. Nevertheless, the extent of performance improvement in comparison to C-CdS/PDMS as the friction layer is not substantial, potentially owing to the limited dispersion effect of N-CdS within the PDMS matrix. In this work, N-CdS demonstrates inferior dispersibility in PDMS compared to C-CdS. This inadequate dispersion is primarily attributed to the tendency of nanomaterials to agglomerate. Finally, as illustrated in Figure 5g, the superior output performance of N-CdS/PDMS verifies the effectiveness of our approach in enhancing TENG performance by inserting N-CdS into the PDMS matrix.

For a comprehensive evaluation of the electrical output performance, we tested the TENG composed of Cu foil and an N-CdS/PDMS membrane at various frequencies and tapping pressures. Figure 6a–c illustrate the variations in output voltage, short-circuit current, and charge transfer density of the TENG across different frequencies. All these parameters exhibited an increasing trend as the frequencies were raised. Specifically, within the frequency range of 0.5 to 2 Hz, the peak-to-peak voltage increased from 57.6 to 236 V, the peak-to-peak short-circuit current increased from 6.4 to 17.4 μA, and the charge transfer density increased from 41 to 55 μm/m^2^, as shown in Figure 6d. Based on the data presented in Figure 6a–d, it is evident that increasing the frequency substantially enhances the output performance, aligning with findings from prior research. The experimental results reaffirm that augmenting the frequency is a viable method for enhancing the performance of TENGs utilizing N-CdS/PDMS as the friction layer. Furthermore, we examined the impact of an applied force on the TENG’s output performance. The results revealed that with an increase in applied pressure from 1 to 5 N, the output voltage increased from 104 to 234 V, as depicted in Figure 6e. This observation indicates that increasing the external pressure effectively boosts the TENG’s output voltage. Similarly, in our investigation of the effect of an external force on the TENG’s output current, as illustrated in Figure 6f, we observed that as the applied pressure rose from 1 to 5 N, the output current increased from 10.2 to 17.4 μA. Thus, it can be inferred that increasing the applied pressure results in a corresponding increase in the output current. Finally, we conducted a study on the impact of an applied force on the charge transfer of the TENG. In contrast to the output voltage and current, our findings indicated that augmenting the applied force has no discernible effect on the charge transfer of the TENG. In TENG studies, it is a common observation that the output power rises with both frequency and applied pressure. Increasing the frequency enhances energy transfer efficiency, consequently raising the maximum output power. Additionally, the increase in pressure leads to a rise in friction force (f = μN, where f represents the friction force, μ is the coefficient of friction, and N is the normal force), thus generating more frictional energy and augmenting the triboelectric output. We demonstrated that the output performance increased tremendously with frequency and applied pressure. Figure 6i illustrates the relationship between the force and output voltage. Increasing the applied force results in the rise of the output voltage of the TENG. In essence, with adjustments to certain parameters and calibration of the instrument, it holds significant promise for its application in mechanical sensing. The TENG’s response to varying forces aligns with trends reported in prior research [37]. Hence, the fabricated TENG has the potential to serve as a force sensor when calibrated against a known force reference.

To explore the maximum output performance of N-CdS/PDMS-based TENGs in practical circuits and facilitate a comparison with C-CdS-PDMS-based TENGs, we conducted tests on their output energy density with an external load resistor connected. We applied load resistance to the circuit using a custom-made resistor box capable of providing resistances ranging from 10 KΩ to 900 MΩ. We then measured the resulting output voltage and current under a force of 5 N at a frequency of 2 Hz. Figure 7a,b display the output voltage and current of the N-CdS/PDMS-based TENG under various external load resistances, respectively. For a clearer comparison, we plotted the peak values of the output voltage and current against different external load resistors in Figure 7c. As shown in Figure 7c, the response of the output voltage and current varies significantly with increasing external load resistance. The output voltage steadily increased with higher external load resistance, reaching 226 V in the end. Notably, the voltage exhibits rapid growth when the load resistance was below 100 MΩ, which was followed by a considerable slowdown beyond this threshold, nearly reaching stagnation. On the contrary, the behavior of the output current is distinctly different. As the external load resistance rises, the output current notably declines, particularly before the 100 MΩ mark, which was followed by a slower rate of decrease beyond this point. The highest output current achieved at 200 KΩ was 23.2 μA. These divergent trends in voltage and current changes are primarily attributed to the circuit voltage and inherent factors of the TENG. The power density exhibited a rapid increase to a maximum value and subsequently decreased (Figure 7d). The peak power of a TENG can be calculated using the following equation:
P_peak_ = UI = U^2^/R(1)
where U is the output voltage, I is the short-circuit current, and R is the load resistance. The calculated P_peak_ values initially rose, which was followed by a decline as R increased. The TENG achieved a maximum power density of approximately 4 W/m^2^ at a load resistance of 9 MΩ. Subsequently, to gain a comprehensive understanding of the impact of small-sized CdS on TENG output performance, we investigated TENGs based on C-CdS/PDMS. As illustrated in Figure 7e, upon the addition of a load resistor, the output voltage of the C-CdS/PDMS-based TENG initially demonstrates rapid growth, eventually stabilizing around 184 V. Similarly, the current exhibits a similar pattern, peaking at 15 μA before rapidly declining and then gradually stabilizing. Although both C-CdS/PDMS and N-CdS/PDMS-based TENGs exhibit analogous changes in output current and voltage following the introduction of a load resistor, the peak voltage and current of the N-CdS/PDMS-based TENG surpass those of the C-CdS/PDMS-based counterpart. This observation further underscores the effective enhancement of TENG performance through modifying the sized of CdS. Furthermore, the maximum output performance of the TENG based on C-CdS/PDMS was calculated, and it is presented in Figure 7f. The results indicate a maximum output energy density of approximately 2.5 W/m^2^ for the TENG based on C-CdS/PDMS, which is less than that of the TENG based on N-CdS/PDMS. This outcome serves as additional evidence supporting the concept that reducing the size of CdS effectively amplifies the output power of the TENG.

We rectified the output voltage of the N-CdS/PDMS-based TENG using a full bridge rectifier, as depicted in Figure 8a. The conversion of the non-rectified output voltage into rectified voltage resulted in a negligible difference, as illustrated in Figure 8b. This suggests that the TENG’s output signals can be efficiently converted into rectified signals without any decrease in performance. To harness the stable power generated by the TENG, we connected LEDs in series, as shown in Figure 8c. Furthermore, the TENG was capable of directly powering 17 LEDs continuously. We also conducted the tests on our device at different time intervals, including 3 days and 5 days, to assess its repeatability, as illustrated in Figure 8d. This demonstrates that TENG based on N-CdS/PDMS has relatively good stability. Our results suggest that CdS holds promise as a potential candidate for cost-effective and high-output energy-harvesting devices.

## 4. Conclusions

In summary, we present a high-performance TENG that integrates nanostructure CdS and PDMS. This cost-effective TENG can efficiently generate stable electrical outputs during periodic mechanical compression and release. Our research demonstrates that the introduction of N-CdS to PDMS significantly enhances the output performance of the TENG in comparison to a TENG based solely on pure PDMS. More precisely, upon the introduction of 0.6 wt% CdS to the PDMS, the CdS/PDMS-based TENG achieved output voltages of 236 V and currents of 17.4 μA, respectively. These values represent a threefold improvement in output voltage and a 2.5-fold improvement in current compared to the original PDMS-based TENG. We also compared the performance to that of the original CdS and found that reducing the size of CdS to the sub-micrometer level effectively enhances the TENG’s output performance. Although the performance enhancement is not substantial, this might be attributed to the poor dispersion characteristics in PDMS, which is likely stemming from the tendency of nanostructured CdS to aggregate. Addressing this issue will be a significant focus for our future research efforts. Furthermore, we investigated the effects of pressure force and frequency on the TENG’s performance, including voltage, current, and charge transfer. We achieved a maximum power density of 4 W/m^2^ under the external load resistance of 9 MΩ, demonstrating TENG’s ability to power LEDs. The results indicate that CdS is a promising candidate for TENGs. However, due to the tendency of CdS nanoparticles to easily agglomerate within the polymer matrix and potentially impact the output performance of the TENG, addressing this issue will be one of our future research priorities.

## Figures and Tables

**Figure 1 nanomaterials-13-02943-f001:**
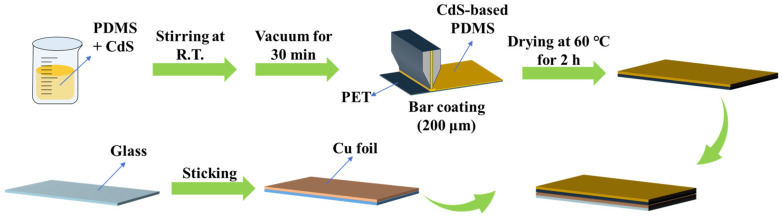
Schematic illustration of the CdS-based TENG fabrication process.

**Figure 2 nanomaterials-13-02943-f002:**
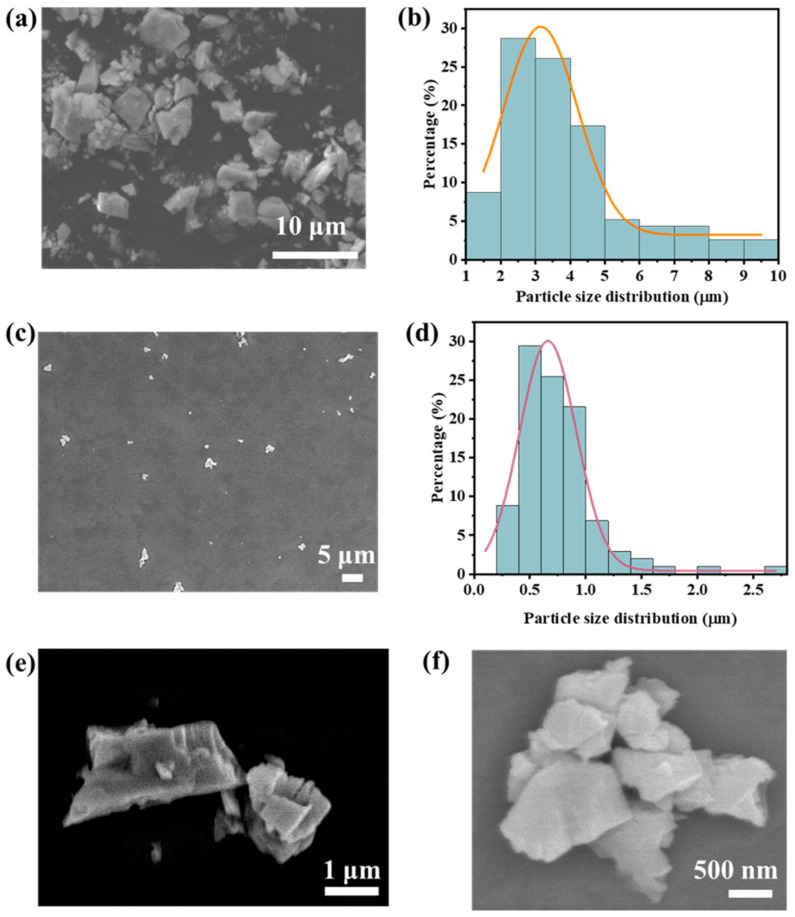
Morphology and size distribution of CdS powder. (**a**) SEM image of C-CdS powder (bar = 10 μm). (**b**) Particle size distribution of C-CdS powder. (**c**) SEM image of N-CdS powder (bar = 5 μm). (**d**) Particle size distribution of N-CdS powder. (**e**,**f**) Morphology of N-CdS power (bar = 1 μm and 500 nm).

**Figure 3 nanomaterials-13-02943-f003:**
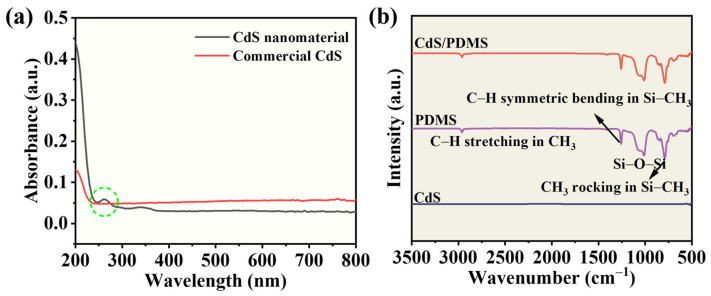
(**a**) UV-visible absorption spectrum for N-CdS and C-CdS dispersion in water. (**b**) FTIR spectra of CdS, PDMS, and CdS/PDMS.

**Figure 4 nanomaterials-13-02943-f004:**
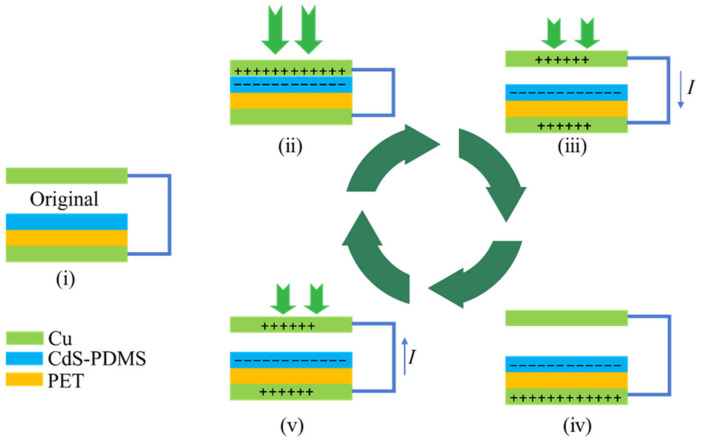
Scheme of working mechanisms and charge generation processes. (i) In initial position with no charge on their faces. (ii) The pressed state. (iii) The releasing processes. (iv) The released state. (v) The pressing processes.

**Figure 5 nanomaterials-13-02943-f005:**
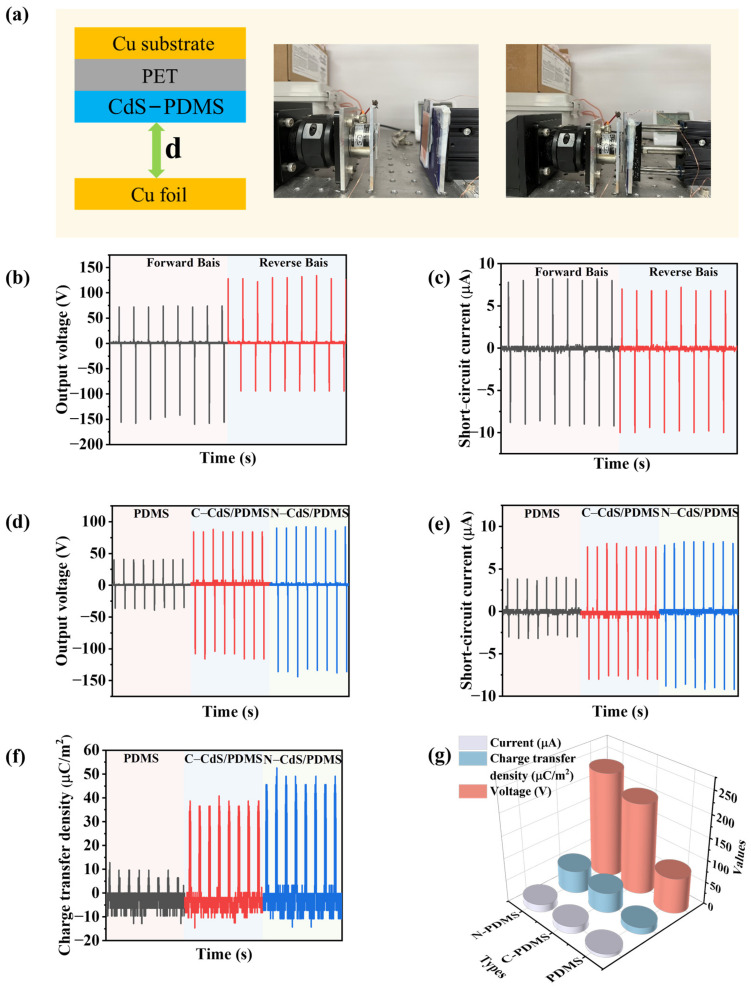
(**a**) Schematic illustration of the CdS-PDMS-based TENG and the image of test setup. (**b**) Forward–reverse open-circuit output voltage characteristics of TENG. (**c**) Forward–reverse short-circuit current of TENG. The output voltage (**d**), current (**e**), and charge transfer density (**f**) signals were harvested from the TENG device with pure PDMS, C-CdS/PDMS, and N-CdS/PDMS. (**g**) Comparison of the output performance values of TENGs based on PDMS, C-CdS/PDMS, and N-CdS/PDMS, including output voltage, current, and charge transfer density.

**Figure 6 nanomaterials-13-02943-f006:**
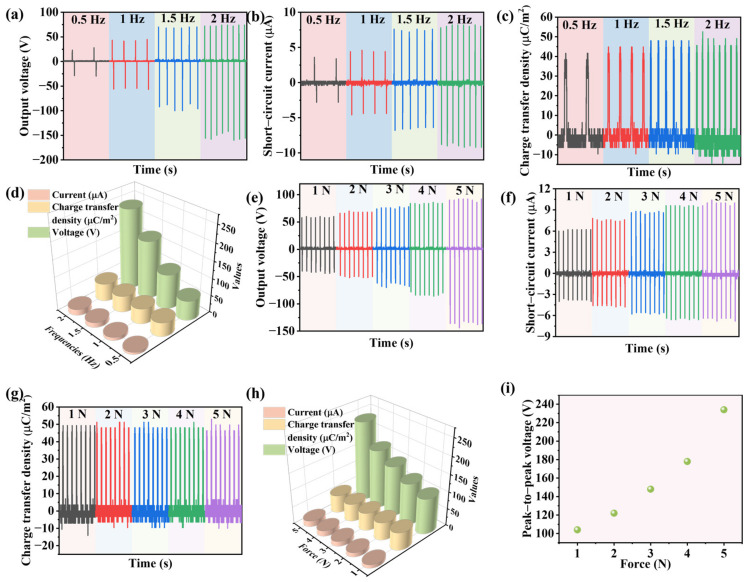
(**a**) Output voltage, (**b**) output current, and (**c**) charge transfer density under loading frequencies of 0.5, 1, 1.5, and 2 Hz. (**d**) Comparison of the output performance values of TENGs based on different loading frequencies, including output voltage, current, and charge transfer density. The output voltage (**e**), output current (**f**), and charge transfer density (**g**) under different loading forces. (**h**) Comparison of the output performance values of TENGs based on different loading forces, including output voltage, current, and charge transfer density. (**i**) Average output voltage as a function of different applied forces.

**Figure 7 nanomaterials-13-02943-f007:**
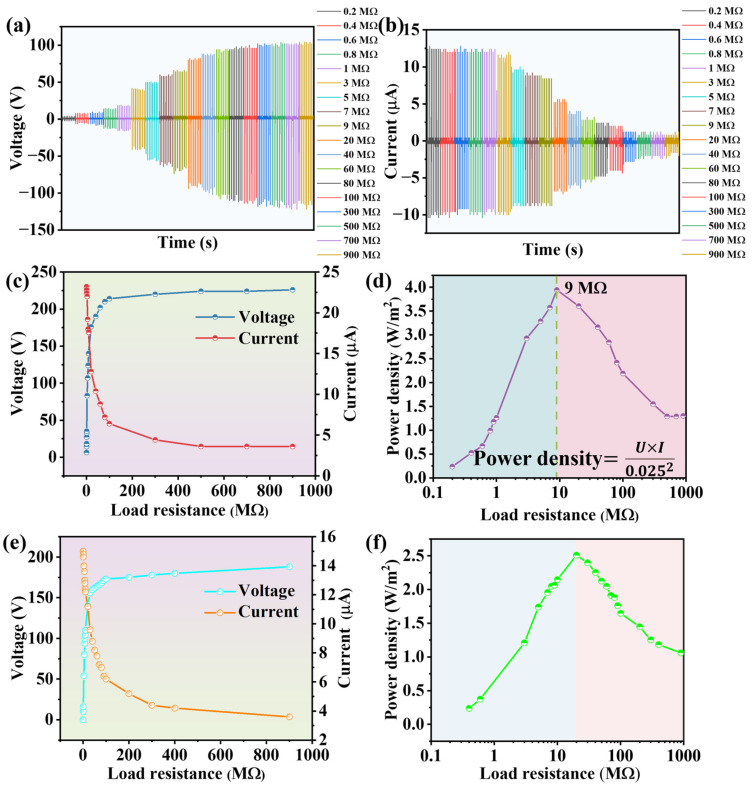
(**a**) Output current and (**b**) output voltage of the N-CdS/PDMS-based TENG as a function of load resistance. (**c**) Under different load resistance, the output current and voltage of the N-CdS/PDMS based TENG. (**d**) Under different load resistance, the output power of the N-CdS/PDMS-based TENG. (**e**) Under different load resistance, the output current and voltage of the C-CdS/PDMS-based TENG. (**f**) Under different load resistance, the output power of the C-CdS/PDMS-based TENG.

**Figure 8 nanomaterials-13-02943-f008:**
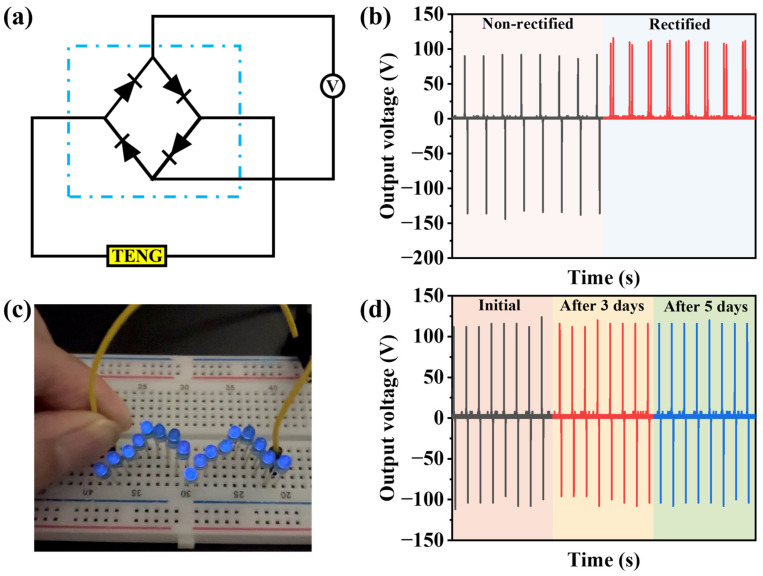
(**a**) Diagram of rectified circuit. (**b**) Rectified output voltage with a full-bridge rectifier. (**c**) Powering blue LEDs with TENG. (**d**) The stability test of TENG over different time intervals.

## Data Availability

Data are contained within the article.

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
