# Peer review of "Improving the Performance of Polydimethylsiloxane-Based Triboelectric Nanogenerators by Introducing CdS Particles into the Polydimethylsiloxane Layer"

_nanomaterials, 2023, doi:10.3390/nano13222943_

Round 1
Reviewer 1 Report
Comments and Suggestions for Authors
Mao and Seo present a comparative study of triboelectric energy harvesting devices using N-CdS/PDMS, commercial CdS/PDMS (C-CdS/PDMS), and pure PDMS substrates. This study is well-conducted and offers interesting results for the scientific community. However, there are a few points that should be further discussed. Therefore, I recommend major revisions before publication in Nanomaterials:
- I suggest providing more details on Figure 1.
- Pleade address the CdS commercial reference within the manuscript. I also consider critical to add XRD analysis of this material or another technique to identify the material.
- In my opinion, the z-axis of Figures 5g, 6d, and 6h should be labeled as "Voltage (V)" instead of "Values," even if the authors have this information as a subtitle.
- Although Figure 8c shows the LEDs ON, this image doesn’t display the entire schematic. I recommend adding a video as supporting information.
- The subtitle of Figure 8d is missing.
- The authors refer to CdS having piezoelectric properties in the introduction, and in the beginning of the conclusions, they mention that they "present a high-performance TENG that integrates piezoelectric CdS and PDMS." However, the possible piezoelectric effect in these devices is not even mentioned in the Results and Discussion section. The authors should analyze the possibility of piezoelectric influence in the results.
Author Response
Response to the REviewer 1
Improving the performance of PDMS-based triboelectric nanogenerators by introducing CdS particles into PDMS layer
Thank you for your decision and constructive comments on my manuscript. The red part in the manuscript that has been revised according to your comments. Revision notes, point-to-point, are given as follows:
We are very grateful to you for reviewing the paper so carefully. We have tried our best to improve the manuscript and have modified something in accordance with your precious suggestion.
Mao and Seo present a comparative study of triboelectric energy harvesting devices using N-CdS/PDMS, commercial CdS/PDMS (C-CdS/PDMS), and pure PDMS substrates. This study is well-conducted and offers interesting results for the scientific community. However, there are a few points that should be further discussed. Therefore, I recommend major revisions before publication in Nanomaterials:
- I suggest providing more details on Figure 1.
Reply: Thank you for your constructive suggestion. According to the Reviewer’s comment, we have provided more experimental details on Figure 1. (Please see the revised manuscript in lines 112-129 of page 3, displayed in red font.)
- Please address the CdS commercial reference within the manuscript. I also consider critical to add XRD analysis of this material or another technique to identify the material.
Reply: Thank you very much for your valuable suggestion. We agree with your opinion. In response to the Reviewer's suggestion, we have included the UV-Vis spectrum data of a commercial CdS sample in our paper to enable a more comprehensive comparison with the nanostructured CdS. The corresponding figure (Figure 3a) has been modified, and the associated content has also been added and revised. Please refer to the revised manuscript, specifically on line 180 of page 5 and line 197 of page 6. Additionally, we agree with your comments that XRD analysis and other techniques for material identification are necessary. Your suggestions have provided guidance for our future research. Unfortunately, the limited availability of XRD equipment within our institution made it challenging to conduct the analysis within the given time. Due to these time constraints, we were unable to supplement XRD characterization data. In our forthcoming studies, we plan to conduct a more comprehensive analysis. Once again, thank you for your constructive input into our work.
- In my opinion, the z-axis of Figures 5g, 6d, and 6h should be labeled as "Voltage (V)" instead of "Values," even if the authors have this information as a subtitle.
Reply: We first apologize for any misunderstanding or confusion caused by our Figures. The data in Figure 5g compares the output performance of TENGs based on different materials (PDMS, C-CdS/PDMS, N-CdS/PDMS). These performance data encompass output voltage, current, and charge transfer density. The value on the Z-axis not only represents voltage data, but also represents the value of current and transferred charge. In Figures 6d, and 6h, we compared the output voltage, current, and charge transfer density data at different frequencies and forces. Additionally, the value on the Z-axis does not solely represent the output voltage, it also represents the current and charge transfer density. The use of “value” instead of “voltage” appears appropriate as it allows for a broader interpretation of the data encompassing multiple parameters. In order to avoid causing confusion to readers, the following content is added to the revised manuscript, “(g) Comparison of the output performance values of TENGs based on PDMS, C-CdS/PDMS, and N-CdS/PDMS, including output voltage, current, and charge transfer density.”, “(d) Comparison of the output performance values of TENGs based on different loading frequencies, including output voltage, current, and charge transfer density.”, “(h) Comparison of the output performance values of TENGs based on different loading forces, including output voltage, current, and charge transfer density.” (Please see the revised manuscript in lines 262-264 of page 9 and in lines 301-307 of page 10, respectively, displayed in red font.)
- Although Figure 8c shows the LEDs ON, this image doesn’t display the entire schematic. I recommend adding a video as supporting information.
Reply: We thank for the Reviewer’s constructive suggestion. According to this comment, we have added a short video as supporting information to display the entire schematic. Please see the provided video.
- The subtitle of Figure 8d is missing.
Reply: We are very sorry for missing the subtitle of Figure 8d. The subtitle has been added as follows “(d) The stability test of TENG over different time intervals.” and it is added at Line 368 of page 13, shown in red font.
- The authors refer to CdS having piezoelectric properties in the introduction, and in the beginning of the conclusions, they mention that they "present a high-performance TENG that integrates piezoelectric CdS and PDMS." However, the possible piezoelectric effect in these devices is not even mentioned in the Results and Discussion section. The authors should analyze the possibility of piezoelectric influence in the results.
Reply: Thank you for the Reviewer’s constructive comments. CdS is a piezoelectric semiconducting material. There are several reports that has introduced the inherent piezoelectric nature of CdS (Phys. Rev. Lett. 4 (10) (1960) 505-507; Appl. Phys. Lett. 92 (2) (2008) 022105; Appl. Phys. Lett. 87 (12) (2005) 123505; Sci. Adv. 2 (7) (2016)). In 2006, Wang et al. reported that they used a piezoelectric nanogenerator (PENG) as an energy harvester based on the piezoelectric effect (Science 312 (2006) 242-246). In addition, anecdotal evidence suggests that nanostructured materials have higher piezoelectric coefficients than their bulk counterparts (Adv. Funct. Mater. 18 (2008) 3553-3567). CdS nanomaterials are also used in energy conversion and harvesting based on the semiconductor and piezoelectric properties of CdS (Sol. Energy Mater. Sol. Cells 90 (2006) 166–174; Thin Solid Films 305 (1997) 124–129). Piezoelectric nanomaterials can convert mechanical energy into electrical energy (Appl. Sci. 2022, 12(14), 7237). When a piezoelectric material is subjected to an applied voltage or mechanical vibration, the induced displacement of ions results in a net electric charge due to a change in the dipole moment of the unit cell, creating a piezoelectric potential across the material (Nature 228, 473–474 (1970), Nat. Mater. 15, 78–84 (2016)). In addition, piezoelectric materials can be manufactured in small sizes and lightweight configurations, making them suitable for integration into compact devices and systems (Nano energy, 80 (2021) 105567). Based on the reported results, we think that piezoelectric CdS also has various possible advantages in TENGs applications. The following content has been added to our manuscript, “Adding CdS, known for its inherent piezoelectric properties, to the TENG could heighten its responsiveness to external variations, including pressure and frequency.” Please see the line 241-243 of page 7, and the added content is displayed in red font).

Reviewer 2 Report
Comments and Suggestions for Authors
The paper is devoted to the current topic of energy harvesting technology using the example of triboelectric nanogenerators and is aimed at improving the characteristics of the active layer of TENG - PDMS using nanostructured CdS. The paper obtained interesting results and its content corresponds to the Nanomaterials profile.
There are some comments.
1. What power of ultrasound was used for processing commercial CdS?
2. The thickness of active PDMs can affect the performance of PDMS-based triboelectric nanogenerators. Studies varying the thickness of the PDMS layer would be useful.
3. The approach of preparing nanostructured CdS starting from CdS microparticles is certainly a simple and economical method and is convenient for scaling up. However, it is known that the top-down method has significant disadvantages, namely, the production of nanoparticles with a wide distribution and often agglomerated nanoparticles. It was possible to make a comparative analysis using monodisperse CdS nanoparticles and with a more uniform distribution of them throughout the polymer matrix.
4. Authors mention mesoporous CdS, but the work does not provide data on the textural characteristics of the nanostructured CdS and resulting CdS/PDMS materials.
5. It would also be useful to estimate the band gap of nanostructured CdS.
The article may be published taking into account the comments made.
Author Response
Response to the REviewer 2
Improving the performance of PDMS-based triboelectric nanogenerators by introducing CdS particles into PDMS layer
We are grateful to you for a careful and thorough review, and for raising important issues that relate directly to the clarity of the manuscript and the interpretation of the data. After discussing your report, we agree with the suggestion of you to submit the revised manuscript to Nanomaterials. We respond to your comments below, point by point, and describe the associated revisions to the manuscript.
The paper is devoted to the current topic of energy harvesting technology using the example of triboelectric nanogenerators and is aimed at improving the characteristics of the active layer of TENG - PDMS using nanostructured CdS. The paper obtained interesting results and its content corresponds to the Nanomaterials profile.
- What power of ultrasound was used for processing commercial CdS?
Reply: In our experiment, the power of ultrasound used for processing commercial CdS is 200 W. The following content has been added, “Additionally, the set power is 200 W.” (Please see the line 107-108 of page 3, and the added content is displayed in red font).
- The thickness of active PDMs can affect the performance of PDMS-based triboelectric nanogenerators. Studies varying the thickness of the PDMS layer would be useful.
Reply: We thank you for your constructive suggestion. We agree with your comments that studies varying the thickness of the PDMS layer would be useful. We conduct research on the relationship between thickness and TENG output performance. The study found (as shown in the figure below) that as the thickness of N-CdS/PDMS increases, the output voltage and current will decrease accordingly, which is consistent with the results of previous research reports (RCS Advances, 2020, 10(30), 17752-17759 It is known that the device performs best when the PDMS thickness is approximately 50 μm. However, due to the limitations of our current fabrication system, we were unable to accurately produce a component with a thickness of less than 200 μm using the bar coating method within the constrained timeframe. Therefore, we were only able to accurately create PDMS thin films with a thickness of 200 μm or more for performance measurement. Additionally, we observed a decrease in device performance for thicknesses exceeding 200 μm, but we did not include the results of this experiment in the main text.
Figure The voltage (a) and short-circuit current (b) of different thickness N-CdS/PDMS based TENGs.
- The approach of preparing nanostructured CdS starting from CdS microparticles is certainly a simple and economical method and is convenient for scaling up. However, it is known that the top-down method has significant disadvantages, namely, the production of nanoparticles with a wide distribution and often agglomerated nanoparticles. It was possible to make a comparative analysis using monodisperse CdS nanoparticles and with a more uniform distribution of them throughout the polymer matrix.
Reply: We appreciate the reviewer’s constructive suggestion. It is simple and economical for us to fabricate CdS nanomaterials by ultrasonication method. However, it is true that the as-prepared nanoparticles have a wide distribution in size, and they often agglomerate together. In our work, compared with the TENG based on a micron structure, the output performance of the TENG based on nanostructured CdS has not shown significant improvement. This could be attributed to the agglomeration of nanostructured CdS in the polymer matrix. We also address this issue in our paper. In our future work, we will dedicate our efforts to resolving the agglomeration of the nanostructured CdS. Based on this situation, in the conclusion part, the following content has been added, “Because the easy agglomeration of CdS nanomaterial in the polymer matrix can impact the output performance of the TENG, addressing this issue will be one of our future research focuses.” (Please see the revised manuscript in lines 387-389 of page 13, displayed in red font.)
- Authors mention mesoporous CdS, but the work does not provide data on the textural characteristics of the nanostructured CdS and resulting CdS/PDMS materials.
Reply: We are very sorry for our incorrect writing, we have deleted the word “mesoporous” in our paper, and it is rectified at Line 138 of page 4.
- It would also be useful to estimate the band gap of nanostructured CdS.
Reply: We thank for your constructive suggestion. We agree that the estimation the band gap of nanostructured CdS would be useful to our work. According to the previous literature, CdS is an n-type semiconductor with a 2.42 eV direct band gap when it is in cubic structure, and 2.57 eV when it is in hexagonal structure (solar energy 157 (2017) 342-348). Currently, we do not have the necessary equipment to perform band gap measurements on the material used in the experiment, making it difficult to estimate the band gap of nanostructured CdS. In our future work, we will estimate the band gap of nanostructured CdS.

Round 2
Reviewer 1 Report
Comments and Suggestions for Authors
Regarding my comment #4, the video should display the experimental setup, meaning the device connected to the LEDs and receiving the mechanical stimulus.
I believe the authors may not have understood my comment 6. TENGs are triboelectric devices, so it is assumed that the only phenomenon generating energy is triboelectric. However, if CdS is a piezoelectric material, it will certainly contribute to the output of the devices. This should be addressed in the Results and Discussion section, especially since it is mentioned in the conclusions without prior discussion.
Author Response
- Regarding my comment #4, the video should display the experimental setup, meaning the device connected to the LEDs and receiving the mechanical stimulus.
Reply: We thank you for your suggestions. According to the Reviewer’s comment, we have uploaded a new video of the device connected to the LEDs and receiving the mechanical stimulus.
- I believe the authors may not have understood my comment 6. TENGs are triboelectric devices, so it is assumed that the only phenomenon generating energy is triboelectric. However, if CdS is a piezoelectric material, it will certainly contribute to the output of the devices. This should be addressed in the Results and Discussion section, especially since it is mentioned in the conclusions without prior discussion.
Reply: We are sorry for misunderstanding the Reviewer’s comments. It is true that a piezoelectric material will certainly contribute to the output performance of the devices. The following content has been added to the manuscript, “CdS is a piezoelectric material, as highlighted in the introduction. Combining the piezoelectric and triboelectric effects can significantly enhance the performance of TENG [36]. When a mechanical force is applied, the triboelectric material PDMS, containing piezoelectric properties of CdS, undergoes deformation, leading to an immediate manifestation of piezoelectricity. The supplementary surface charges resulting from the piezoelectric effect of CdS reinforce the triboelectric charge during simultaneous contact electrification, ultimately boosting the device's electrical outputs.” Please see the line 241-248 of page 7, and the added content is displayed in red font.
